# Generic Cutting Force Modeling with Comprehensively Considering Tool Edge Radius, Tool Flank Wear and Tool Runout in Micro-End Milling

**DOI:** 10.3390/mi13111805

**Published:** 2022-10-22

**Authors:** Shuaishuai Gao, Xianyin Duan, Kunpeng Zhu, Yu Zhang

**Affiliations:** 1School of Mathematics and Statistics, Central China Normal University, Wuhan 430079, China; 2Key Laboratory of Metallurgical Equipment and Control Technology, Ministry of Education, Wuhan University of Science and Technology, Wuhan 430081, China; 3Institute of Intelligent Machines, Hefei Institutes of Physical Science, Chinese Academy of Sciences, Hefei 230031, China

**Keywords:** cutting force, mechanical modeling, micro-end milling, tool edge radius, tool flank wear, tool runout

## Abstract

Accurate cutting force prediction is crucial in improving machining precision and surface quality in the micro-milling process, in which tool wear and runout are essential factors. A generic analytic cutting force model considering the effect of tool edge radius on tool flank wear and tool runout in the micro-end milling process is proposed. Based on the analytic modeling of the cutting part of the cutting edge in the end face of the micro-end mill bottom, the actual radius model of the worn tool is established, considering the tool edge radius and tool flank wear. The tool edge radius, tool wear, tool runout, trochoidal trajectories of the current cutting edge, and all cutting edges in the previous cycle are comprehensively considered in the instantaneous uncut chip thickness calculation and the cutter–workpiece engagement determination. The cutting force coefficient model including tool wear is established. A series of milling experiments are performed to verify the accuracy and effectiveness of the proposed cutting force model. The results show that the predicted cutting forces are in good agreement with the experimental cutting forces, and it is necessary to consider tool wear in the micro-milling force modeling. The results indicate that tool wear has a significant influence on the cutting forces and cutting force coefficients in the three directions, and the influences of tool wear on the axial cutting force and axial force coefficient are the largest, respectively. The proposed cutting force model can contribute to real-time machining process monitoring, cutting parameters optimization and ensuring machining quality.

## 1. Introduction

With the fast-growing demand for miniature components and products in microelectronics, optics, aerospace, and industry, micro-manufacturing technology has been studied widely and promoted greatly in recent years [1,2,3,4]. Micro-milling is one of the most important and productive micro-manufacturing technologies. It plays a major role in manufacturing complex feature parts due to the advantage of flexibility and high accuracy in machining a variety of metals materials [1,2,3,4].

In the micro-milling process, the machining feature is within the range of 1 nm~1 mm, and the diameter of the micro-mill is below 1 mm [2]. As the tool and machining feature size decrease sharply, the tool edge radius is close to, or even greater than, the instantaneous uncut chip thickness (IUCT). Micro-milling is distinct from traditional milling in machining mechanisms, such as size effect [5], minimum uncut chip thickness [6], and tool runout [7]. These factors accelerate the tool wear, resulting in poor product surface quality and low machining accuracy. The milling force is one of the most direct and important signals in the micro-milling process. Mechanical cutting force modeling in micro-milling is important for cutting force prediction and optimizing machining parameters. It contributes to raising machining quality and production efficiency, improving the cutting state of the tool, extending the tool’s life, and reducing machining costs. Much of the present research involves cutting force modeling.

Most of the cutting force modeling has adopted the mechanical model [8]. The main work involves undeformed cutting thickness modeling, cutter–workpiece engagement (CWE) determination, and cutting force coefficient identification. In the micro-end milling process, the instantaneous uncut chip thickness (IUCT) modeling considered the trochoidal trajectories of the tool tip [9,10], tool runout [7,11], tool deflection [12], elastic recovery [13], and the chip thickness accumulation phenomenon [14]. Zhang et al. [15] comprehensively considered several factors and proposed a force model, including the tool runout of the axial offset and axial tilt in the micro-end milling process. In addition to the undeformed cutting thickness modeling, CWE determination and cutting force coefficient identification also have an impact on the cutting force prediction. In the micro-end milling process, the entry and exit angles calculation has considered the tool runout [7,16], the minimum uncut chip thickness [12], and tool deflection [15]. Some research has calibrated the cutting force coefficients using the orthogonal cutting finite element method considering the tool edge radius [17,18,19,20,21]. Zhang et al. [15] proposed an analytic cutting force model in micro-end milling, considering three material removal mechanisms. Wojciechowski et al. [14] applied a dual approach based on the finite element method, and measured instantaneous cutting force in the cutting force coefficients calibration. Zhang et al. [16] included size effect in the cutting force coefficients.

Due to the small diameter of the micro-milling tool, high spindle speed, and size effect caused by the cutting edge radius and chip thickness accumulation, the tool wears rapidly in micro-milling. Instantaneous cutting forces increase with tool wear. Establishing a mechanical cutting force model considering tool wear is essential in the micro-milling process. Various force models were proposed to compensate for the increased force generated by tool wear. The effect of tool wear on the cutting force in the macro-machining process can be used for reference in the micro-milling process. Shao et al. [22] proposed an analytic milling force model considering average tool flank wear and cutting conditions in conventional end-face-milling. Sun et al. [23] presented a mathematical cutting force model, considering the relationship between tool flank wear and the cutting force in end-milling. Chinchanikar and Choudhury [24,25] established a cutting force model of oblique cutting, considering the effect of tool wear in the turning process. The cutting forces include the cutting forces of the sharp tool and the rubbing forces of the tool flank wear land. Hou et al. [26] put forward a mechanical cutting force model considering tool wear based on the stress distribution of tool flank wear. Orra and Choudhury [27] proposed a mechanistic cutting force model in the turning process. The cutting force included the forces due to chip formation, tool nose edge radius, and tool wear-out. Nouri et al. [28] described a new method of real-time monitoring of end-mill wear by tracking force coefficients independent of cutting conditions during end milling. It used an indirect approach to evaluate tool wear in the macro-machining process. Shi et al. [29] used the polynomial approximation method to model the cutting forces considering tool wear in end-milling.

Some researchers have presented the mechanical cutting force models considering tool wear in the micro-milling process. Bao and Tansel [30] suggested the tool wear effect in the cutting force model by wear coefficients that were estimated by using a genetic algorithm. Oliaei and Karpat [31] analyzed the influence of tool wear on milling force coefficients in the micro-milling process by combining the finite element method and experiment method. Results showed that the cutting forces are related to the tool wear types, and tool flank wear is dominant in milling. Lu et al. [32] obtained tool flank wear using the finite element method, and then established the micro-milling force model considering tool flank wear. Results showed that tool flank wear significantly impacted the cutting forces. Zhang et al. [33] proposed a cutting force model considering tool runout and random tool wear considering the cutting edge trochoidal trajectories, and tool runout with axial offset and tilt. A particle filter algorithm was used to estimate the random tool wear. Li et al. [34] expressed the cutting force model considering tool runout and tool flank wear. The forces produced by tool flank wear were modeled as increments of the cutting force coefficients on the original model. Liu et al. [35] represented a cutting force model including variable cutting parameters, tool runout, and tool wear. Zhu and Li [36] presented a cutting force model of IUCT considering tool runout and tool wear, and the cutting force coefficients of each cutting edge were established as exponential functions of each shrinkage of the tooth radius. Liu et al. [37] established a cutting force model comprehensively considering tool runout, size effect, and tool wear. The nonlinear shear and ploughing coefficient models including cutting edge radius, and a friction coefficient model considering tool flank wear are constructed, respectively.

The tool tip was simplified as a sharp tip when analyzing tool wear in the cutting force model without considering tool edge radius. The diameter of the micro-mill is small, the feed rate is low, and the IUCT and tool edge radius are of the same magnitude order in the micro-milling process. The effect of the tool edge radius cannot be ignored in micro-milling. Tool flank wear is the primary type of tool wear in the micro-milling process. There are some cutting force models based on tool flank wear in the micro-milling process. It is necessary to take tool flank wear considering tool edge radius (VBr) into account in the cutting force model. To accurately predict the cutting forces in the micro-end milling process, we consider VBr and tool runout in the cutting force modeling. The innovation is modeling the actual radius of the worn tool considering VBr and tool runout, based on the analytic model of the cutting part of the end face of the micro-end mill bottom. The IUCT is calculated and the CWE is determined by comprehensively considering VBr and tool runout, the trochoidal trajectories of the current cutting edge and all passing cutting edges in the previous cycle. The cutting force coefficients of the worn tool are established considering tool flank wear.

## 2. Generic Cutting Force Modeling in the Micro-End Milling Process

The main work of the paper is shown in Figure 1. Firstly, a series of micro-end milling experiments were designed and carried out to obtain the milling forces and the corresponding tool flank wear. Next, the actual radius of the worn tool considering VBr is analyzed. Based on this, CWE and IUCT are determined, respectively. Afterwards, cutting force coefficients are calibrated with the measured cutting forces and tool flank wear. Then, cutting forces are predicted by adopting the proposed model. The tool runout is calibrated with the measured forces and the presented cutting force model, and fed back to the modeling of CWE, IUCT and the cutting force coefficient.

### 2.1. Generic Cutting Force Model in Micro-End Milling

To model the cutting forces, the geometric relationship in the micro-end milling process is introduced in Figure 2. A typical micro-end mill with two teeth is used. The Cartesian coordinate system *O-xyz* is set up, as shown in Figure 2a. The center *O* of the end face of the micro-end mill bottom is the origin of the coordinate system. The *x*-axis is along the feed direction of the tool, and the *z*-axis is upward along the tool axis. The *y*-axis is perpendicular to both the *x*-axis and *z*-axis, as shown in Figure 2a. Based on the method of the mechanical cutting force model [8], the micro-end mill is divided into a series of equal elements slices (Figure 1) of equal height along the *z*-axis direction (Figure 1a). Each tool element is considered to be orthogonal cutting at a small inclination angle (Figure 2b). Tangential element cutting force d*F_ti_*(*θ*) and radial element cutting force d*F_ri_*(*θ*), axial element cutting force d*F_ai_*(*θ*) of the point *P* at the specific axial cutting depth *z* of *i*th (*i* = 0, 1, 2, …, *N* − 1), *N* is the cutting edge numbers of the tool, *i* is the ordinal number of cutting edge, and the ordinal number of tool teeth is set counterclockwise. The cutting edge can be described as [17]:(1)$$\{\begin{array}{c} dF_{ti}(\theta )=K_{tc}h(\theta )dz,\\ \begin{array}{l} dF_{ri}(\theta )=K_{rc}h(\theta )dz,\\ dF_{ai}(\theta )=K_{ac}h(\theta )dz, \end{array} \end{array}$$
where *K_tc_*, *K_rc_*, and *K_ac_* are the actual tangential, radial, and axial directions cutting coefficients, respectively; *h*(*θ*) is the instantaneous uncut chip thickness, *θ* is the immersion angle of the element, i.e., the included angle counterclockwise from the positive direction of the *y*-axis to the direction of the tool center *O_i_* to the point *P*, and:(2)$$\theta (i,t,z)=\omega t-\frac{2\pi i}{N}-\frac{z\tan\beta }{R_{a}},$$
where *R_a_* is the actual radius of the worn tool, *β* is the helix angle of the tool, *ω* is angular velocity of the spindle(rad/s), and *ω* = 2π*n*/60, *n* is the spindle speed(r/min), *t* is the time(s), and d*z* is the axial cutting depth of the disk element. Referring to the geometric condition in Figure 2a, d*z* is given as:(3)$$dz=\frac{R_{a}d\theta }{\tan\beta },$$
where d*θ* is the radial element immersion angle as shown in Figure 2b.

Considering the geometric condition in Figure 2b, the tangential, radial, and axial element cutting forces are transformed into cutting forces in the *x*-, *y*- and *z*-axes direction by the following equations:(4)$$\left(\begin{array}{c} dF_{xi}(\theta )\\ \begin{array}{l} dF_{yi}(\theta )\\ dF_{zi}(\theta ) \end{array} \end{array}\right)=\left(\begin{array}{ccc} -\cos\theta & -sin\theta & 0\\ sin\theta & -\cos\theta & 0\\ 0 & 0 & 1 \end{array}\right)\left(\begin{array}{c} dF_{ti}(\theta )\\ \begin{array}{l} dF_{ri}(\theta )\\ dF_{ai}(\theta ) \end{array} \end{array}\right).$$

Integrate both sides of the above equation with *θ* over the CWE, and then sum with the ordinal numbers *i* of the cutting edge, by using Equations (1)–(4). The total cutting force of the tool at the rotation angle *φ* = *ωt* is obtained:(5)$$\left(\begin{array}{c} F_{x}(\varphi )\\ \begin{array}{l} F_{y}(\varphi )\\ F_{z}(\varphi ) \end{array} \end{array}\right)=\frac{R_{a}}{\tan\beta }\left(\begin{array}{ccc} -\sum_{i=0}^{N-1}\int_{\theta_{l}}^{\theta_{u}}\cos\theta h(\theta )d\theta & -\sum_{i=0}^{N-1}\int_{\theta_{l}}^{\theta_{u}}\sin\theta h(\theta )d\theta & 0\\ \sum_{i=0}^{N-1}\int_{\theta_{l}}^{\theta_{u}}\sin\theta h(\theta )d\theta & -\sum_{i=0}^{N-1}\int_{\theta_{l}}^{\theta_{u}}\cos\theta h(\theta )d\theta & 0\\ 0 & 0 & \sum_{i=0}^{N-1}\int_{\theta_{l}}^{\theta_{u}}h(\theta )d\theta \end{array}\right)\left(\begin{array}{c} K_{tc}\\ \begin{array}{l} K_{rc}\\ K_{ac} \end{array} \end{array}\right),$$
where *θ_u_* and *θ_l_* are the upper and lower limits of the cutting edge immersion angle, respectively; i.e., the corresponding immersion angles of the points at the bottom, and the maximum cutting depth position of the cutting edge at any given tool rotation angle. According to the cutting edge trochoidal trajectories, we can calculate the entry and exit angles, and then determine *θ_u_* and *θ_l_*.

Sorting Equation (5), a generic analytic model of the cutting force considering VBr and tool runout in the micro-end milling process, can be obtained as follows:(6)$$\left(\begin{array}{c} F_{x}(\varphi )\\ \begin{array}{l} F_{y}(\varphi )\\ F_{z}(\varphi ) \end{array} \end{array}\right)=\frac{R_{a}}{\tan\beta }\left(\begin{array}{c} \sum_{i=0}^{N-1}\int_{\theta_{l}}^{\theta_{u}}\left(-K_{tc}\cos\theta -K_{rc}\sin\theta \right)h(\theta )d\theta \\ \sum_{i=0}^{N-1}\int_{\theta_{l}}^{\theta_{u}}\left(K_{tc}\sin\theta -K_{rc}\cos\theta \right)h(\theta )d\theta \\ \sum_{i=0}^{N-1}\int_{\theta_{l}}^{\theta_{u}}K_{ac}h(\theta )d\theta \end{array}\right).$$

### 2.2. Actual Tool Radius Modeling Considering Tool Edge Radius and Tool Flank Wear

To accurately model the cutting force in the micro-end milling process, the actual worn tool radius model considering VBr will be built.

The tool tip coordinate system $x_{E}O_{E}y_{E}$ considering the tool edge radius is established in the end face of the micro-end mill bottom, as shown in Figure 3 and Figure 4. In the tool tip coordinate system $x_{E}O_{E}y_{E}$, the straight line that passes through the center of the cutter and tangent to the arc of the tool tip is the *y_E_*-axis, and the direction from the tangent point to the cutter center is the positive direction of the *y_E_*-axis. The straight line perpendicular to the *y_E_*-axis and tangent to the arc of the tip is the *x_E_*-axis. The intersection *O_E_* of the *x_E_*-axis and the *y_E_*-axis is the tool tip coordinate system origin, as shown in Figure 3 and Figure 4. Figure 4 is the diagram of VBr and the shrinkage of tooth radius. Line segment *CL*, arc *CAH* and line segment *HL* are the rake face, cutting edge arc, and flank face of the cutting edge of the micro-end mill bottom, respectively. *VB* is tool flank wear, and shrinkage of the tooth radius Δ*R* is *OW* in Figure 3 and Figure 4. The ideal tool tip coordinate system $x_{E}^{\prime }O_{E}^{\prime }y_{E}^{\prime }$ is established in the end face of micro-end mill bottom, as shown in Figure 4. The ideal tool tip $O_{E}^{\prime }$ is the coordinate system origin, the direction from point $O_{E}^{\prime }$ to the cutter center is the positive direction of the $y_{E}^{\prime }$-axis, and the $x_{E}^{\prime }$-axis is perpendicular to the $y_{E}^{\prime }$- axis. In the ideal coordinate system $x_{E}^{\prime }O_{E}^{\prime }y_{E}^{\prime }$, tool flank wear *V’B’* and shrinkage of the tooth radius Δ*R’*, i.e., $O_{E}^{\prime }U$ are bigger than the actual values respectively in Figure 4. In the coordinate system $x_{E}O_{E}y_{E}$, the coordinates of points *A*, *C*, *D* and *H* are as follows:A(0,r),C(r−rcosγ,r+rsinγ),D(r,0),H(r+rsinα,r−rcosα),
where *r* is the tool edge radius, *γ* and *α* are the tool rake angle and the tool clearance angle, respectively.

The actual cutting part in the end face of the micro-end mill bottom is described as:(7)$$x=\{\begin{array}{lll} y\tan\gamma +r\left(1-\tan\gamma -\sec\gamma \right), & y\in [\;r+r\sin\gamma ,\Delta R^{*}], & \\ r-\sqrt{r^{2}-{\left(y-r\right)}^{2}}, & y\in [0,r+r\sin\gamma ), & \\ r+\sqrt{r^{2}-{\left(y-r\right)}^{2}}, & y\in [0,r-r\cos\alpha ), & \\ y\cot\alpha +r\left(-\cot\alpha +\csc\alpha +1\right), & y\in [r-r\cos\alpha ,\Delta R^{*}], & \end{array}$$
where Δ*R** is the critical value of Δ*R* corresponding to the maximum tool flank wear *VB*^*^. Generally, *VB** is a given value according to the actual machining requirements.

Tool wear changes dynamically, and tool flank wear gradually increases with the cutting length. Tool flank wear goes through three periods, as shown in Figure 4a–c. According to the geometric relationship shown in Figure 4 and Equation (7), the relationship model between tool flank wear *VB* and shrinkage of the tooth radius Δ*R* is established as follows:(8)$$\begin{array}{ll} VB & =x(B)-x(V)\\ & =\{\begin{array}{ll} 2\sqrt{r^{2}-{\left(r-\Delta R\right)}^{2}}, & \Delta R\in [0,r-r\cos\alpha ),\\ \Delta R\cot\alpha +r\left(-\cot\alpha +\csc\alpha \right)+\sqrt{r^{2}-{\left(r-\Delta R\right)}^{2}}, & \Delta R\in [r-r\cos\alpha ,r+r\sin\gamma ),\\ \Delta R\left(\cot\alpha -\tan\gamma \right)+r\left(-\cot\alpha +\csc\alpha +\tan\gamma +\sec\gamma \right), & \Delta R\in [r+r\sin\gamma ,\Delta R^{*}]. \end{array} \end{array}$$

Generally, tool flank wear reaches the threshold *VB*^*^ at the stage in Figure 4c. According to Equation (8), we can obtain:(9)$$\Delta R^{*}=\frac{VB^{*}-r\left(-\cot\alpha +\csc\alpha +\tan\gamma +\sec\gamma \right)}{cot\alpha -\tan\gamma }.$$

It can be easily proved that function *VB* is strictly monotonically increasing with Δ*R* on the interval [0, Δ*R**]. Hence, *VB* as a function of the variable is invertible on the interval [0, Δ*R**].

Consequently, according to Equation (8), the mathematical model for the shrinkage radius of the worn tool Δ*R* of tool flank wear *VB* in the coordinate system of the actual tool tip coordinate system can be established. According to the geometric relationship in Figure 3, the actual radius of the worn tool is described as:(10)$$\begin{array}{ll} R_{a} & =R-\Delta R\\ & =\{\begin{array}{ll} R-r+\sqrt{r^{2}-\frac{VB^{2}}{4}} & VB\in \left[0,\;2r\sin\alpha \right),\\ R-r+r\cos\alpha -VB\cos\alpha \sin\alpha +\sin^{2}\alpha \sqrt{\Delta }, & VB\in \left[2r\sin\alpha ,\;r\left(1+\csc\alpha \right)\right),\\ R-r+r\cos\alpha -VB\cos\alpha \sin\alpha -\sin^{2}\alpha \sqrt{\Delta }, & VB\in \left[\;r\left(1+\csc\alpha \right),\;r\left(\csc\alpha +\cos\gamma +\cot\alpha \sin\gamma \right)\right),\\ R-\frac{VB+r\left(\cot\alpha -\csc\alpha -\tan\gamma -\sec\gamma \right)}{\cot\alpha -\tan\gamma }, & VB\in \left[r\left(\csc\alpha +\cos\gamma +\cot\alpha \sin\gamma \right),\;VB^{*}\right], \end{array} \end{array}$$
where $\Delta ={\left[VB\cot\alpha +r(\cot^{2}\alpha -\cot\alpha \csc\alpha +1)\right]}^{2}-\sec^{2}\alpha {\left[VB+r(\cot\alpha -\csc\alpha )\right]}^{2}$, *R* is the fresh tool radius. *R_a_* is strictly monotonically increasing with *VB* on the interval [0, *VB**]. Therefore, when *VB*ϵ[0, *VB**], there is a unique *R_a_* corresponding to each *VB*.

### 2.3. IUCT Modeling Considering Tool Edge Radius, Tool Flank Wear, and Tool Runout

The geometry of tool runout is shown in Figure 5. The tool runout distance *r*_0_ is the distance between the ideal centerline, and the actual centerline of the micro-end mill. The tool runout angle *λ* is the angle from the positive direction of *y*-axis to the direction from the first tool tip, to the offset direction at the end face of the tool bottom. It is specified that the tool runout angle is positive when it is clockwise, and negative when it is counterclockwise.

Accurate IUCT modeling is essential for cutting force prediction in the micro-end milling process. The position of the micro-end mill center changes with the tool runout. The tool radius reduces with tool wear. Thus, the cutting edge trochoidal trajectories change.

The IUCT is calculated by comprehensively considering VBr, tool runout, the trochoidal trajectories of the current cutting edge and all passing cutting edges in the previous cycle. According to the cutting edge trochoidal trajectories, the IUCT of the point *P_i_* on the *i*th cutting edge at the position angle *θ_i_* = (*ωt_i_* − 2*πi*/*N* − *z*tan*β/R_a_*) corresponding to time *t_i_* and can be determined as follows:(11)$$\begin{array}{l} h=\left|P_{i}P_{i-j}\right|\\ \;\;\;=\sqrt{\tan^{2}\theta_{i}+1}|R_{a}\cos\left(\omega t_{i}-\frac{2\pi i}{N}-\frac{z\tan\beta }{R_{a}}\right)+r_{0}\cos(\omega t_{i}+\lambda )\\ \;\;\;\;\;\;\;\;-R_{a}\cos\left(\omega t_{i-j}-\frac{2\pi (i-j)}{N}-\frac{z\tan\beta }{R_{a}}\right)-r_{0}\cos(\omega t_{i-j}+\lambda )|,\\ \;\;(\theta_{i}\neq \frac{\pi }{2}+2m\pi ,m\in N,N\;\mathrm{is}\;\mathrm{the}\;\mathrm{natural}\;\mathrm{numbers}\;\mathrm{set}), \end{array}$$
or
(12)$$\begin{array}{l} h=\left|P_{i}P_{i-j}\right|\\ \;\;\;=|R_{a}sin\left(\omega t_{i}-\frac{2\pi i}{N}-\frac{z\tan\beta }{R_{a}}\right)+\frac{ft_{i}}{60}+r_{0}\sin(\omega t_{i}+\lambda )\\ \;\;-R_{a}sin\left(\omega t_{i-j}-\frac{2\pi (i-j)}{N}-\frac{z\tan\beta }{R_{a}}\right)-\frac{ft_{i-j}}{60}-r_{0}\sin(\omega t_{i-j}+\lambda )|,\\ \;\;(\theta_{i}=\frac{\pi }{2}+2m\pi ,m\in N,N\;\mathrm{is}\;\mathrm{the}\;\mathrm{natural}\;\mathrm{numbers}\;\mathrm{set}), \end{array}$$
where point *P_i−j_* is on the (*i−j*)th cutting edge corresponding to time *t_i−j_*, *f* is the feed rate (mm/min), and *f_z_* is feed per tooth(μm/tooth). Time *t_i_* is a given value, and *t_i−j_* can be obtained by using the Newton iteration method according to the cutting edge trochoidal trajectories. Thus, the IUCT *h* can be determined.

### 2.4. Cutting Force Coefficient Determination Considering Tool Flank Wear

With rapid tool wears, instantaneous cutting forces gradually increase in the micro-end milling process. Tool geometry changes with tool wear, leading to a change in the cutting force coefficients. Tool flank wear is the main type of tool wear. Consequently, the cutting force coefficients are affected by tool flank wear. To accurately predict the cutting forces, tool flank wear must be considered in the cutting force coefficient modeling.

The increased forces caused by tool wear are present in the form of increments in each cutting force coefficient in the cutting force model. Given this, it is assumed that the increments in the cutting force coefficients caused by tool flank wear are in the form of a quadratic polynomial over *VB*. Then the cutting force coefficients considering tool flank wear are as follows:(13)$$\{\begin{array}{c} K_{tc}=k_{tc}+a_{tc}VB^{2}+b_{tc}VB+c_{tc},\\ K_{rc}=k_{rc}+a_{rc}VB^{2}+b_{rc}VB+c_{rc},\\ K_{ac}=k_{ac}+a_{ac}VB^{2}+b_{ac}VB+c_{ac}, \end{array}$$
where *K_tc_*, *K_rc_*, and *K_ac_* are the cutting force coefficients in tangential, radial, and axial directions of a fresh tool, respectively, *a_tc_*, *b_tc_*, *c_tc_*, *a_rc_*, *b_rc_*, *c_rc_*, *a_ac_*, *b_ac_*, and *c_ac_* are constant parameters determined by fitting.

## 3. Experimental Verification and Discussion

### 3.1. Experimental Setup

To verify the accuracy of the proposed model, a series of micro slot end-milling experiments were carried out on the high-speed machining center MIKRON HSM600U (Mikron Group, Agno, Switzerland). The detailed experimental setup is shown in Figure 6. A two teeth cemented carbide micro end mill is used. The geometric parameters of the tool are shown in Table 1. The workpiece material is AISI4340 alloy structural steel, and the workpiece size is 30 mm × 60 mm × 25 mm. During the experiments, the cutting forces and tool wear are measured. A Kistler9119 3-component dynamometer is used to measure the cutting forces in the *x*-, *y*- and *z*-axis direction, and the sampling frequency is 24 kHz. An industrial camera is used to photograph the end face of the micro-end mill bottom to measure tool wear.

Taking the machining parameters as the three factors, the orthogonal experiments of three factors and three levels are designed by using the full slot milling method. Detailed cutting parameters are shown in Table 2. A fresh tool was replaced for each group of experiments. To reduce the influence of the tool performance differences, the tools used in each group of experiments are all of the same brand, type, and batch. Tool wear was measured per 60 mm of cutting length, and 10 times in each group of experiments, that is, the total cutting length of each tool is 10 times 60 mm. The total cutting of 9 group experiments are 9 × 10 times.

To verify the proposed cutting force model, it is necessary to measure tool flank wear first. Tool flank wear *VB* is directly measured by the measuring tool from the images taken by the industrial camera. As mentioned above, a total of 90 times of cuttings were carried out. After each cutting, the same method was used to measure the tool wear, i.e., 90 times the measurement. Figure 7 shows the tool wear images of a tool before cutting, after the first, sixth, and tenth slot milling in the micro-milling experiment No.4. Considering that the two edges wear values are different and caused by tool runout, the average value of two cutting edge wear is taken as tool flank wear.

### 3.2. Verification and Analysis of Cutting Force Model

#### 3.2.1. Tool Runout Calibration

By marking the tool holder, tool runout parameters are ensured to be the same in each group experiment. According to the measured forces, the calculation steps are as follows, shown in Figure 8.

Step 1: According to the sampling frequency, the tool rotation angles *φ* in one cycle of each group experiment during slot milling are taken as the sample points, and the corresponding experimental value of the milling forces *F_x_* and *F_y_* are obtained.

Step 2: The cutting force coefficients *k_tc_* and *k_rc_* without considering runout are calculated by calibrating the average cutting forces.

Step 3: Tool runout calibration:

(a) The initial and maximum value of the tool runout parameters are set to (*r*_0min_, *λ*_min_) and (*r*_0max_, *λ*_max_), respectively, and the iteration step is set to (Δ*r*_0_, Δ*λ*).

(b) The upper and lower limits of the engaged angle and the IUCT of each cutting edge at each rotation angle corresponding to each group of the tool runout value are calculated. The cutting forces in the *x-* and *y-* axis directions are then simulated.

(c) The square sum *δ*(*r*_0_, *λ*) of the deviations between the simulated forces and the measured forces at each cutting edge sample points in the *x*- and *y*-axis directions corresponding to each group of the tool runout value is calculated.

(d) The tool runout value is calibrated by calculating $\underset{\left(r_{0},\lambda \right)}{\arg}\min\delta (r_{0},\lambda )$, i.e., the group of the tool runout value makes *δ*(*r*_0_, *λ*) the smallest.

Calibration results of the tool runout are shown in Table 3.

#### 3.2.2. The Method of Cutting Force Prediction

Cutting forces are predicted according to the flow chart shown in Figure 9. The steps are as follows:

Step 1: The actual radius of the worn tool *R_a_* is calculated according to the measured flank wear *VB* and Equation (10);

Step 2: The angle upper and lower limits *θ_u_* and *θ_l_* of the engaged cutting edge at each rotation angle;

Step 3: According to Equations (11) and (12), the IUCT of each cutting edge at each rotation angle are calculated;

Step 4: The cutting force coefficients *K_tc_*, *K_rc_*, and *K_ac_* are calculated by using the force model Equation (5) and fitted by using Equation (13).

Step 5: Using the above steps 2–4 and Equation (6), the theoretical cutting forces at each rotation angle are calculated.

#### 3.2.3. Statistical Analysis

To verify the validity and accuracy of the proposed model, four indexes, namely, the mean force errors *E*, Mean Absolute Error *MAE*, Root Mean Square Error *RMSE* and the determination coefficient of *R*^2^ are used to evaluate the model.

In the 9 groups (Table 2) of the micro-end milling experiments, each group is 10 times cutting. To highlight the impact of tool wear, we take the last milling process, i.e., the 10th cutting in each group of experiments as an example to examine the consistency between the predicted and measured cutting forces. The comparison results are shown in Figure 10, Figure 11 and Figure 12 and Table 4.

Figure 10 shows the error *E* between mean forces of the simulated and the measured in the *x*-, *y*-, and *z*-axis directions when the tool wear is considered or not. The results show that when tool wear is considered, the predicted mean force errors in the *x*- and *y*-axis directions are both less than 15%, and the error of the predicted average force in the *z*-axis direction is less than 17%. Thus, the predicted mean forces agree well with the experimental results. It shows that the established model is reliable. Compared with the not considered tool wear, the errors of the predicted average forces in the *x*-axis, *y*-axis, and *z*-axis directions are reduced by more than 25%, 27%, and 58%, respectively. It shows that tool wear significantly affects the forces in the three directions, and the influence on the *z*-axis directional force is the largest. The end face wear of the tool bottom in the *z*-axis direction increases with milling length, increasing the cutting thickness in the *z*-axis direction, so that the *z*-axis directional force is significantly increased. The predicted mean force errors considering tool wear are smaller than without tool wear, which indicates that it is necessary to consider tool wear in the milling force modeling.

Figure 11 shows the Mean Absolute Error *MAE* of the simulated forces with and without tool wear. *MAE* of the simulation forces considering tool wear in the *x*-, *y*-, and *z*-axis directions are over 0.12*N*, 0.22*N*, and 0.15*N* less than that without considering tool wear, respectively. This shows that the established model is effective, and it is necessary to consider tool wear in the milling force modeling.

Figure 12 shows the Root Mean Square Error *RMSE* of the simulation force when the tool wear is considered or not. The *RMSE* of the simulation forces considering tool wear in the *x*-, *y*- and *z*-axis directions are reduced by over 0.6*N*, 1.4*N*, and 0.28*N*, respectively, compared with the not considered tool wear. This shows that the model established is effective. It also shows the necessity of taking tool wear into account in the milling force model.

Table 4 shows the determination coefficient of *R*^2^ of the milling force model. The *R*^2^ for forces in the *x*-axis and *y*-axis directions are greater than 0.7, which shows that the milling force model established in this paper fits the force in the *x*- and *y*-axis directions well.

### 3.3. Discussion

To study the effect of tool wear on the cutting forces in the micro-end milling process, experiment No.4 is taken as an example.

#### 3.3.1. Effect of Tool Wear on the Cutting Force

Table 5 shows that tool flank wear rapidly increases to 37.10 μm after the first cutting. Tool flank wear increases continuously with the cutting times. After the 10th cutting, tool flank wear increased to nearly twice the tool flank wear after the first cutting.

The results of the 1st and 10th cutting of test No.4 are discussed, as shown in Figure 13. It can be seen from Figure 13a,b that tool wear has a small impact on the cutting forces and tool flank wear does not have a significant effect on the results, owing to the fact that tool wear is minimal during the 1st cutting. Figure 13b–d shows that the cutting force peaks in the *x*-axis, *y*-axis, and *z*-axis directions are increased by 48.2%, 48.39%, and 209%, respectively after the 10th cutting, compared with after the 1st cutting. Tool wear causes the cutting forces of the two edges to increase, tool wear has a pronounced influence on the forces in three directions, and the impact on the *z*-axis directional force is the largest. The predicted cutting forces considering tool wear greatly agree with the experimental value. Therefore, establishing a cutting force model considering tool wear is necessary.

#### 3.3.2. Effect of Tool Wear on the Cutting Force Coefficient

According to the milling forces and tool flank wear that were measured by experiments at different stages, the cutting force coefficients at different stages can be obtained by using Equation (5). Using Equation (13), *a_tc_*, *b_tc_*, *c_tc_*, *a_rc_*, *b_rc_*, *c_rc_*, *a_ac_*, *b_ac_*, and *c_ac_* are fitted by the Least Square Method. It can be seen from Figure 14 that the cutting force coefficients increase with tool flank wear. After the 10th cutting, the cutting force coefficients increase by 57.75%, 55.85%, and 495.39% in the *x*-axis, *y*-axis, and *z*-axis directions, respectively. It shows that tool wear has a pronounced effect on the force coefficients in the three directions, and the influence on the *z*-axis directional force coefficient is the largest. Owing to the fact that the axial wear of the end face at the tool bottom increases with the milling length, it rapidly increases the axial cutting forces and the axial force coefficients. It shows that the accurate evaluation of the cutting force coefficient at different stages of tool wear has a critical impact on the cutting force prediction.

## 4. Conclusions

Considering VBr and tool runout, the trochoidal trajectories of the current cutting edge and all passing cutting edges in the previous cycle, a generic cutting force model in the micro-end milling process is proposed. The advantage of the proposed model is the actual radius of the worn tool, considering VBr is established. VBr and tool runout are considered in the calculation of CWE and IUCT. The cutting force coefficients of the worn tool are modeled as functions of tool flank wear. Based on the work done, the following conclusions are drawn:(1)The developed model can dynamically consider the impact of tool runout and tool wear, and can reflect in real-time the cutting state and wear of tools in the cutting process. It is helpful to deepen the research on the micro-milling mechanism and optimize the cutting process.(2)Through a series of micro slot end milling experiments, we have found that the proposed model is more accurate than without considering tool wear. It is necessary to consider the tool wear in the cutting force model based on tool runout. It is due to the gradual increase of the cutting force with the tool wear.(3)Results show that tool wear significantly affects the forces in the three directions, and the influence on the axial force is the largest. This is caused by the axial wear of the tool end face.(4)The results show that the influence of tool wear on the cutting force coefficients in the three directions is very significant, and the axial force coefficient increases by 495.39%. It is significant to establish an accurate cutting force coefficient model considering tool wear. This is due to the increase of axial force caused by tool wear.(5)The established model can be generalized to nano-diamond machining. The research will be extended to different types of tools to establish a more general force model in micro-milling.

## Figures and Tables

**Figure 1 micromachines-13-01805-f001:**
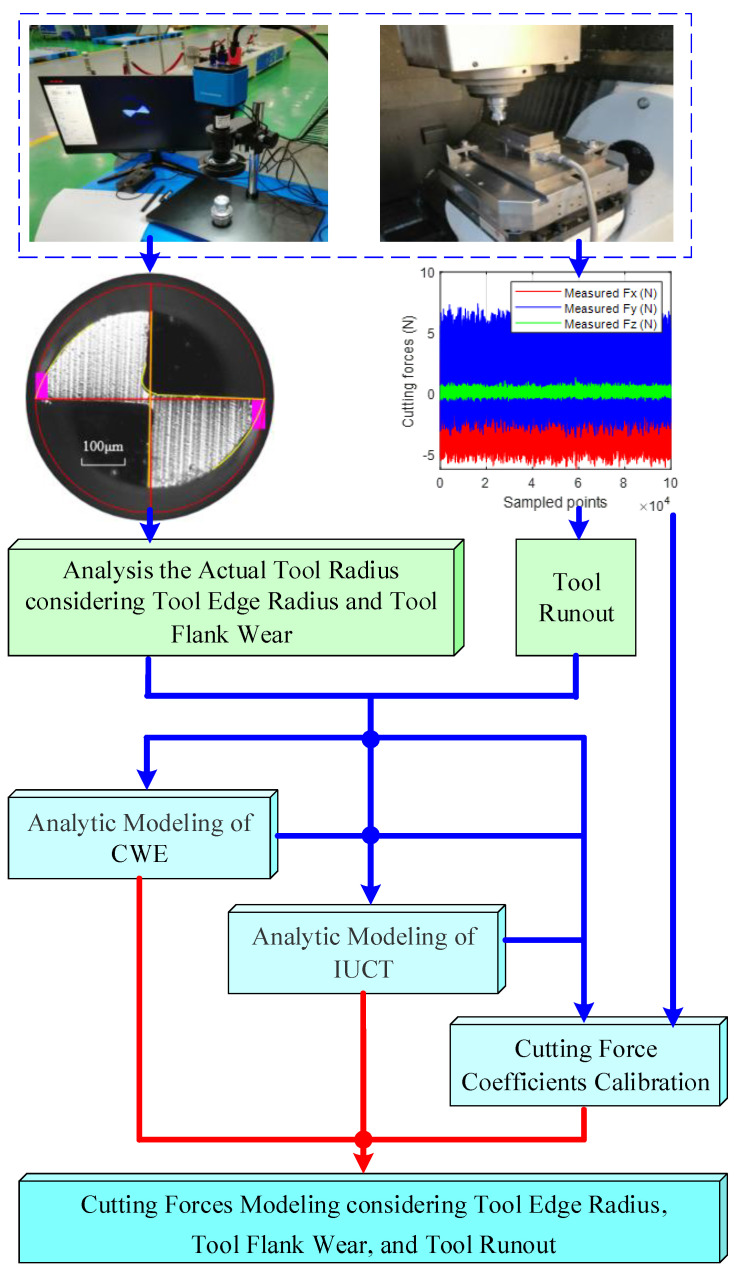
Framework of the generic analytic cutting force model.

**Figure 2 micromachines-13-01805-f002:**
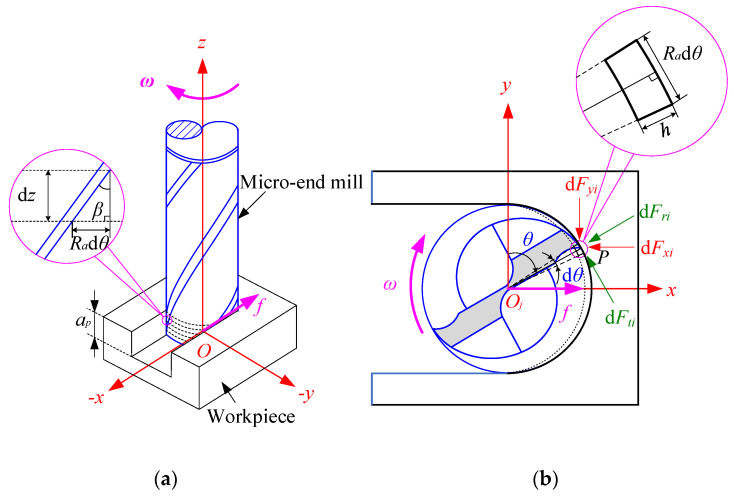
Diagram of the micro-end milling progress. (**a**) Geometry and coordinate system. (**b**) Chip element and element cutting forces.

**Figure 3 micromachines-13-01805-f003:**
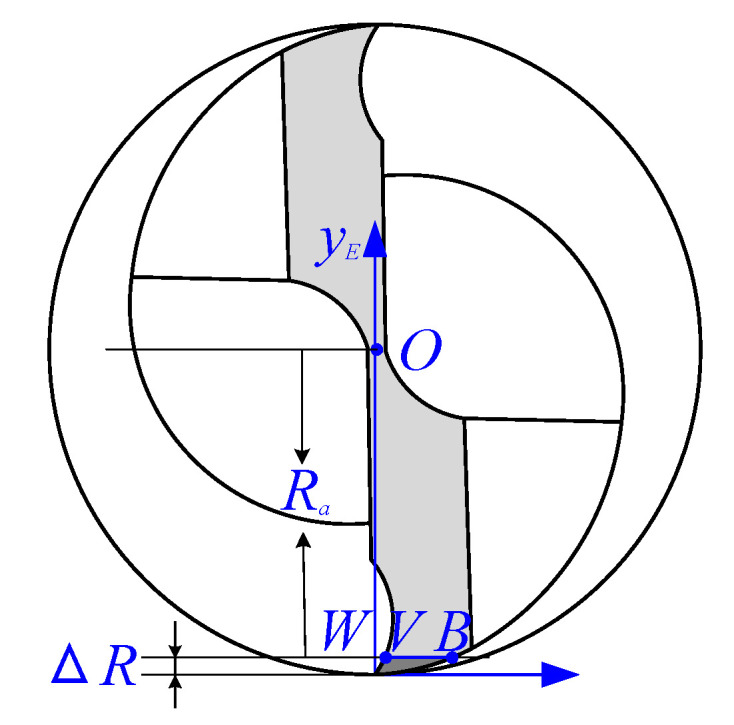
Tool tip coordinate system.

**Figure 4 micromachines-13-01805-f004:**
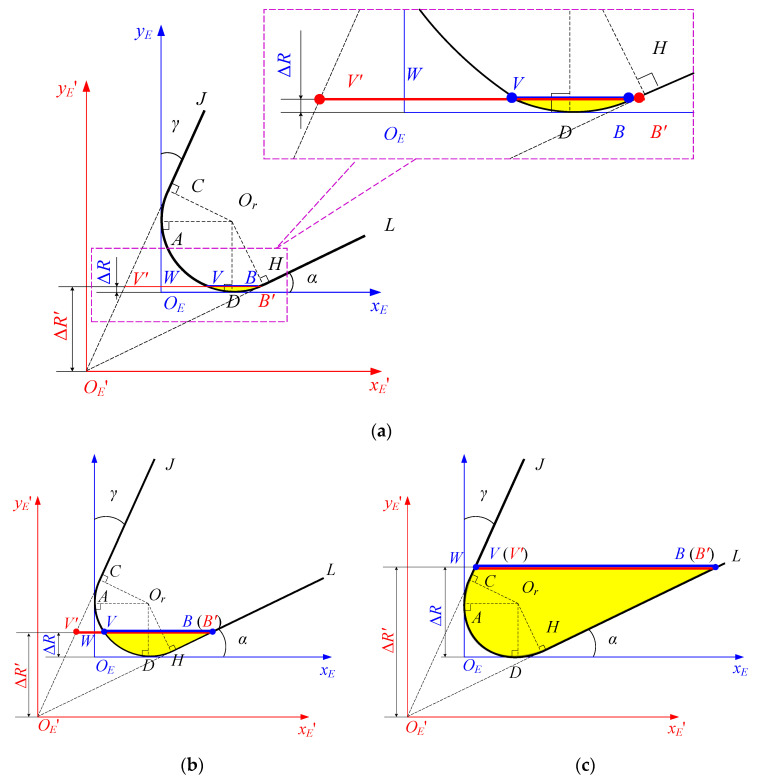
The shrinkage of tooth radius considering VBr. (**a**) 0≤ΔR<r−rcosα; (**b**) r−rcosα≤ΔR<r+rsinγ; (**c**) r+rsinγ≤ΔR<R.

**Figure 5 micromachines-13-01805-f005:**
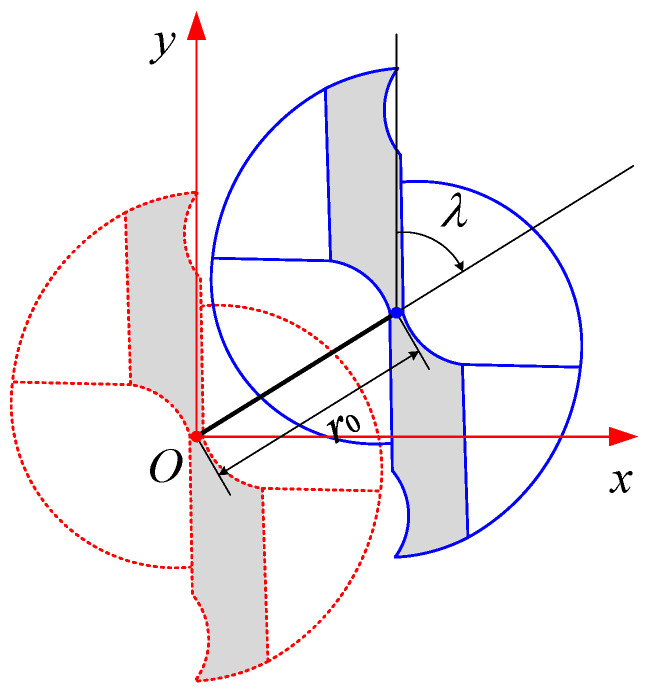
Geometry of the tool runout.

**Figure 6 micromachines-13-01805-f006:**
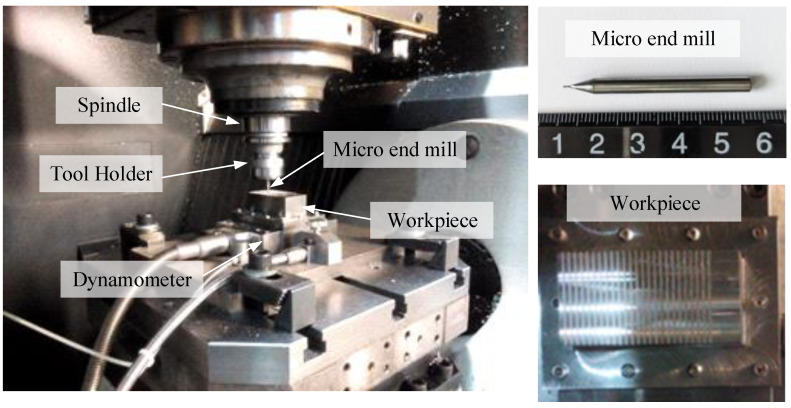
Experimental setup for micro-end milling.

**Figure 7 micromachines-13-01805-f007:**
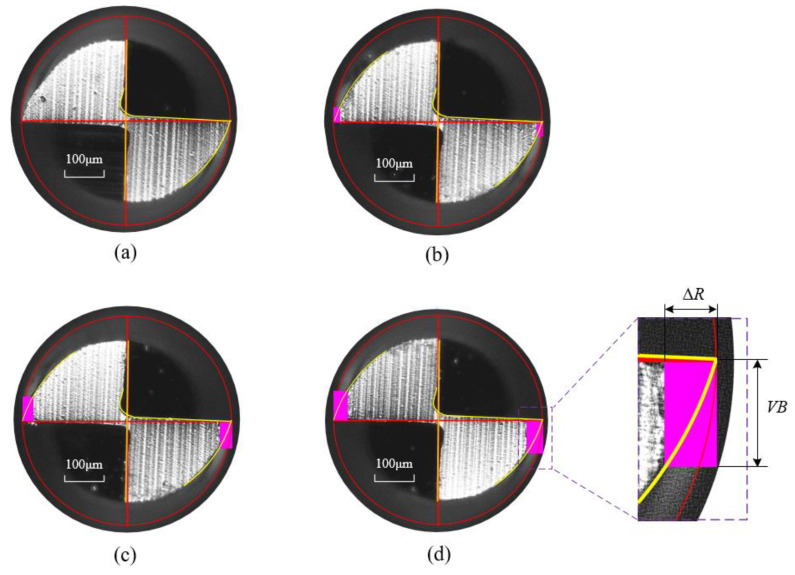
Diagram of tool flank wear in the micro-end milling process. (**a**) Fresh tool; (**b**) After 1st cutting; (**c**) After 6th cutting; (**d**) After 10th cutting.

**Figure 8 micromachines-13-01805-f008:**
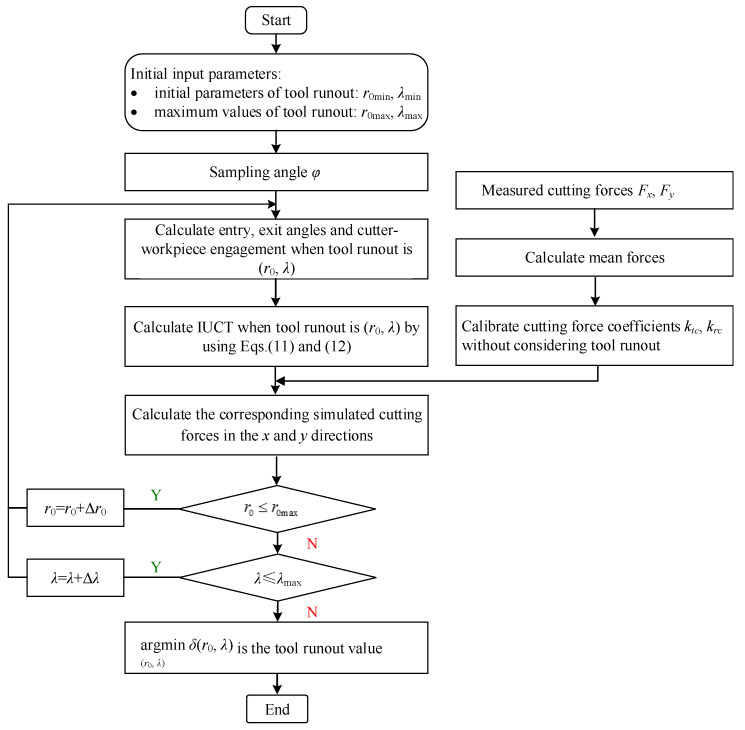
Calculation method of tool runout parameters.

**Figure 9 micromachines-13-01805-f009:**
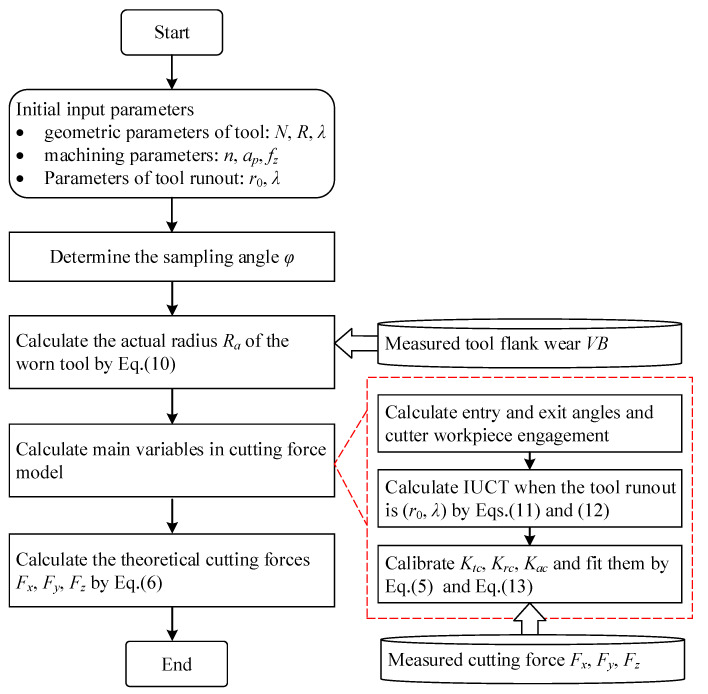
Flow chart of cutting forces prediction in micro-end milling progress.

**Figure 10 micromachines-13-01805-f010:**
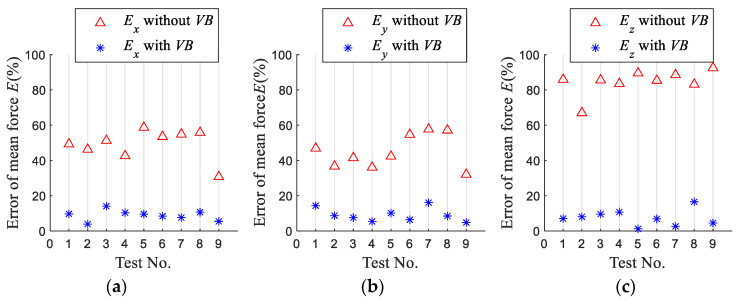
Errors of simulated mean force. (**a**) Mean force error of *F_x_* (**b**) Mean force error of *F_y_* (**c**) Mean force error of *F_z_*.

**Figure 11 micromachines-13-01805-f011:**
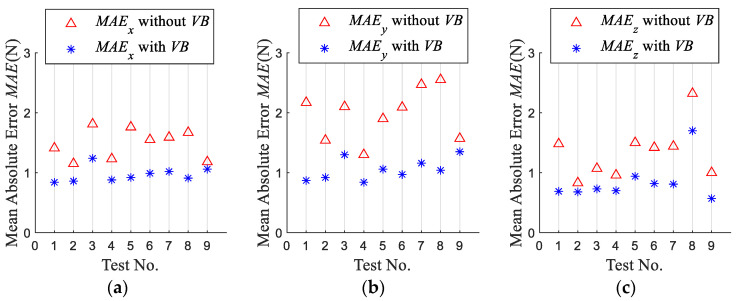
Mean Absolute Error (*MAE*) of simulated force. (**a**) *MAE* of simulated *F_x_* (**b**) *MAE* of simulated *F_y_* (**c**) *MAE* of simulated *F_z_*.

**Figure 12 micromachines-13-01805-f012:**
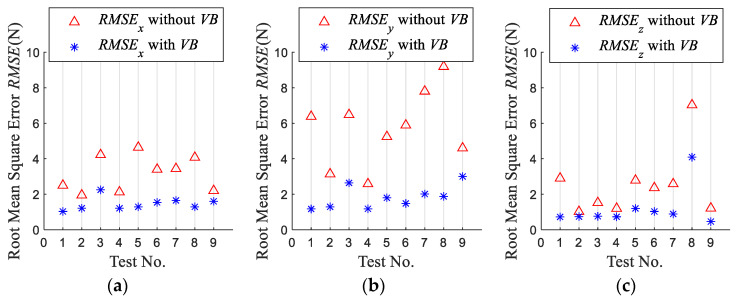
Root Mean Square Error (*RMSE*) of simulated force. (**a**) *RMSE* of simulated *F_x_* (**b**) *RMSE* of simulated *F_y_* (**c**) *RMSE* of simulated *F_z_*.

**Figure 13 micromachines-13-01805-f013:**
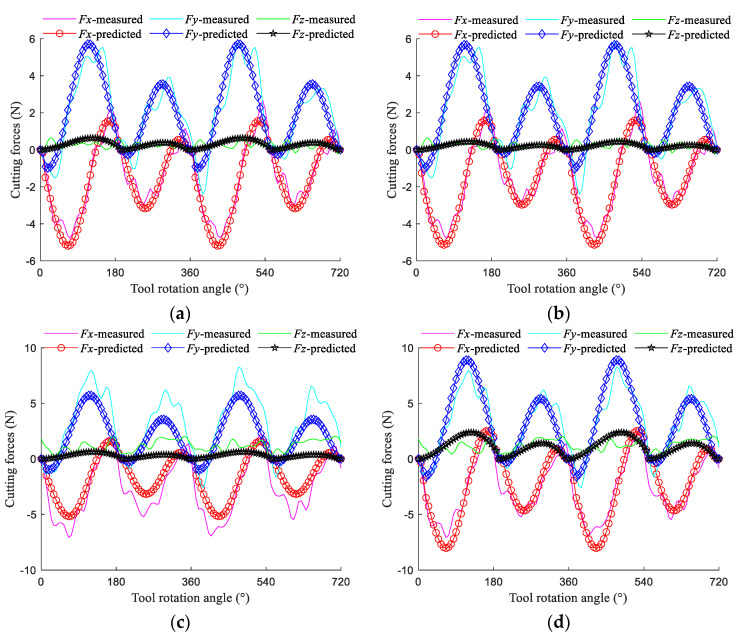
The 1st cutting and 10th cutting in test No.4. (**a**) The 1st cutting without tool wear (**b**) The 1st cutting with tool wear (**c**) The 10th cutting without tool wear (**d**) The 10th cutting with tool wear.

**Figure 14 micromachines-13-01805-f014:**
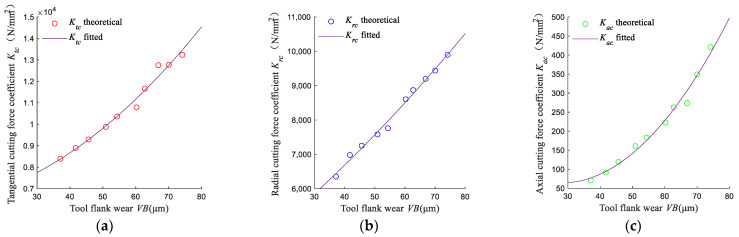
Fitting of cutting force coefficient in No.4. (**a**) Fitting of tangential cutting force coefficients; (**b**) Fitting of radial cutting force coefficients; (**c**) Fitting of axial cutting force coefficients.

**Table 1 micromachines-13-01805-t001:** Geometric parameters of micro-end mill.

Parameter	Value
Tool diameter *d*	0.5 mm
Helix angle *β*	30°
Rake angle *γ*	10°
Clearance angle *α*	5°
Tool edge radius *r*	2 μm

**Table 2 micromachines-13-01805-t002:** Cutting parameters of micro-milling experiments.

Test No.	Spindle Speed *n* (r/min)	Axial Depth of Cut *a*_p_ (μm)	Feed Per Tooth *f*_z_ (μm/Tooth)
1	18,000	60	2
2	18,000	80	4
3	18,000	100	6
4	24,000	80	6
5	30,000	60	6
6	24,000	60	4
7	24,000	100	2
8	30,000	80	2
9	30,000	100	4

**Table 3 micromachines-13-01805-t003:** Results of tool runout.

No.	*r*_0_ (μm)	*λ* (deg)	No.	*r*_0_ (μm)	*λ* (deg)
1	1.531	75	6	1.617	70
2	1.423	−85	7	0.22	−55
3	0.761	76	8	0.671	90
4	1.805	61	9	0.336	85
5	0.378	31			

**Table 4 micromachines-13-01805-t004:** Determination coefficients *R*^2^ of simulated forces.

No.	*R* ^2^ * _x_ *	*R* ^2^ * _y_ *
1	0.79	0.77
2	0.83	0.80
3	0.75	0.76
4	0.82	0.84
5	0.84	0.82
6	0.76	0.81
7	0.72	0.76
8	0.85	0.80
9	0.84	0.75

**Table 5 micromachines-13-01805-t005:** Tool flank wear measured in test No.4.

No.	*VB*	No.	*VB*
1	37.10	6	60.23
2	41.74	7	62.75
3	45.65	8	66.85
4	50.90	9	70.03
5	54.35	10	74.15

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
