# Peer review of "Generic Cutting Force Modeling with Comprehensively Considering Tool Edge Radius, Tool Flank Wear and Tool Runout in Micro-End Milling"

_micromachines, 2022, doi:10.3390/mi13111805_

Round 1
Reviewer 1 Report
I can recommend the publication of this manuscript after a minor revision.
Write keywords in alphabetical order.
The writing must be improved, it is still very poor with numerous wrong terms, typos, or grammar mistakes.
Line 51: minor mistake “...cutter-workpiece engagement(CWE) ....”, and so on.
There is no reference in the text on ref. [21].
Line 81: “Orra et al. [27]” must be written as “Orra and Choudhury [27]”.
Line 89: “Bao et al.” must be written as “Bao and Tansel....”.
Line 91: “Oliaei et al” must be written as “Oliaei and Karpat....”.
Line 99: “Li et al.” must be written as “Li and Zhu....”.
Line 103: “Zhu et al.” must be written as “Zhu and Li....”.
Line 106: minor mistake “Liu et al.[37]...”
Line 116: minor mistake “..tool edge radius(VBr).....”
Fig. 1 – Could you enlarge the pictures from fig. 1 (the upper part)? I can’t distinguish all details.
Line 140: minor mistake “..Model......”
Insert references for all mathematical formulas.
Line 217: minor mistake “Error! Reference source not found.,"
There is no reference in the text to Fig. 8.
Specify the limits of this study.
Insert Authors’ contribution, Institutional Review Board Statement, Informed Consent Statement, Acknowledgements.
Even though the work is relevant to the journal's scope, i.e., Micromachines, I do not find even a single article published in the journal in the list of references.
References are not written according to the Guide of Authors. Verify them and edit them correspondingly. Sometimes are used abbreviations and sometimes are not used (ref. [34] Int J Adv 535 Manuf Technol; ref. [35] The International Journal of Advanced Manufacturing Technology), and so on.
Authors may consider citing the following references:
[1] Corina Bîrleanu, Ştefan Ţălu, Machine elements. Designing and computer assisted graphical representations, Cluj-Napoca, Victor Melenti Publishing house, 2001, 335 p., ISBN 973-99539-6-4.4.
[2] Ş. Ţălu, Micro and nanoscale characterization of three-dimensional surfaces. Basics and applications. Napoca Star Publishing House, Cluj-Napoca, Romania, 2015. ISBN 978-606-690-349-3.
This paper can be published after the mentioned revisions.
Reviewer 2 Report
After reading the manuscript, my comments and suggestions are below;
1. The existing title need modification. No proper reflection of the word “comprehensively”.
2. Minor corrections, like line 41-42 (appropriate position of “,” and “and”.
3. Some recent studies are in the field, while using the cutting tool shape in rock fragmentation application.
a. Comparison Study on Coarseness Index and Maximum Diameter of Rock Fragments by Linear Cutting Tests
b. Analysis of the Effect of the Tool Shape on the Performance of Pre-Cutting Machines during Tunneling Using Linear Cutting Tests
4. In my opinions, no need for the paper organization (last paragraph of the introduction).
5. Equation numbering is missing.
6. While using the symbols, using the insert option for symbols to in line with the text i.e. line 222-228, 252-258, and so on.
7. Include a Figure, to show the different parameters of the tool (table 1).
8. Revise the abstract and be specific at this stage. The abstract is appropriate, however can be concise further.
Reviewer 3 Report
1) Check the complete correctness of the text (eg: in line 116 words are joined; in line 217 there is an Error in the .pdf document, etc.).
2) In Figure 8, there are no markers to continue the flow of the algorithm in both logic loops. It may be implied that the direction down is True (T) and the direction to the left is False (F), but I think it is better to specify the indicated labels.
3) In Figure 9 below the fourth level of blocks, the algorithm branches in three directions without a conditional command and a logical loop (operation). I think that this is not logically possible and that the authors should introduce an additional block with a conditional command and branching in all three directions. If all three horizontal blocks in the fifth level are to be executed then they should simply be arranged one below the other. If the authors did not understand my suggestion and remark, I can help them and draw this algorithm.
